# Impact of Endocytosis Mechanisms for the Receptors Targeted by the Currently Approved Antibody-Drug Conjugates (ADCs)—A Necessity for Future ADC Research and Development

**DOI:** 10.3390/ph14070674

**Published:** 2021-07-15

**Authors:** Manar Hammood, Andrew W. Craig, Jeffrey V. Leyton

**Affiliations:** 1Departément de Medécine Nucléaire et Radiobiologie, Faculté de Medécine et des Sciences de la Santé, Centre Hospitalier Universitaire de Sherbrooke (CHUS), Université de Sherbrooke, Sherbrooke, QC J1H 5N4, Canada; manar.hammood@usherbrooke.ca; 2Department of Biomedical and Molecular Sciences, Queen’s University, Kingston, ON K7L 3N6, Canada; ac15@queensu.ca; 3Centre d’Imagerie Moleculaire, Centre de Recherche, CHUS, Sherbrooke, QC J1H 5N4, Canada

**Keywords:** endocytosis, antibody-drug conjugate, CD33, CD30, CD22, CD79b, BCMA, HER2, nectin-4, trop2, endophilin A2

## Abstract

Biologically-based therapies increasingly rely on the endocytic cycle of internalization and exocytosis of target receptors for cancer therapies. However, receptor trafficking pathways (endosomal sorting (recycling, lysosome localization) and lateral membrane movement) are often dysfunctional in cancer. Antibody-drug conjugates (ADCs) have revitalized the concept of targeted chemotherapy by coupling inhibitory antibodies to cytotoxic payloads. Significant advances in ADC technology and format, and target biology have hastened the FDA approval of nine ADCs (four since 2019). Although the links between aberrant endocytic machinery and cancer are emerging, the impact of dysregulated internalization processes of ADC targets and response rates or resistance have not been well studied. This is despite the reliance on ADC uptake and trafficking to lysosomes for linker cleavage and payload release. In this review, we describe what is known about all the target antigens for the currently approved ADCs. Specifically, internalization efficiency and relevant intracellular sorting activities are described for each receptor under normal processes, and when complexed to an ADC. In addition, we discuss aberrant endocytic processes that have been directly linked to preclinical ADC resistance mechanisms. The implications of endocytosis in regard to therapeutic effectiveness in the clinic are also described. Unexpectedly, information on endocytosis is scarce (absent for two receptors). Moreover, much of what is known about endocytosis is not in the context of receptor-ADC/antibody complexes. This review provides a deeper understanding of the pertinent principles of receptor endocytosis for the currently approved ADCs.

## 1. Introduction

Cancer is the second leading cause of mortality worldwide, resulting in >1 million deaths annually. The unmet medical need for more effective anticancer therapeutics, especially for strategies that can localize cytotoxicity to tumor cells and reduce toxicity to normal tissues, has led to many years of development and ultimately successful realization of antibody-drug conjugates (ADCs), designed to provide the selective elimination of antigen-positive cells. At the time of this review, nine ADCs have been approved by the U.S. Food and Drug Administration for the treatment of six types of blood cancers and two types of solid tumors (Table 1). However, as there are approximately 300 ADCs in clinical trials, there will certainly be more approvals in the near future [1].

This recent success is perhaps a turning point in the field and a strong indication that the time for ADCs may have finally arrived. From 2000 to 2018, only five ADCs showed sufficient efficacy to gain approval. Just one, T-DM1, approved in 2013 for human epidermal growth factor receptor 2 (HER2)-positive breast cancer was approved for the treatment of solid tumors. ADCs are engineered to target overexpressed receptors on the surface of tumor cells and then release cytotoxins that deliver up to 10,000 times the potency of traditional chemotherapy within tumors, while minimizing toxicity to healthy tissues. The approach has been challenged for decades by the difficulty in the selection of a cytotoxic payload, the method of conjugation, and pairing with a monoclonal antibody (mAb) specific for the tumor-associated antigen. Several ADCs were abandoned, as their potency was too weak to effectively kill tumors or too harmful to healthy tissues. However, the ADC field is now in exciting times, as four new ADCs have been approved since 2019 [2,3,4,5]. In addition, three of these ADCs are for the treatment of solid tumors. This recent success is largely due to the ability of scientists to devise improved methods to conjugate ultratoxic payloads to mAbs that enable ADCs to deliver the cytotoxin with improved efficiency within tumor cells and limit premature release during circulation. Essentially, the interaction of the linker type, payload potency and mechanism of action, and biology of the target must all be harmoniously balanced to produce an effective agent.

Apart from the preferential tumor overexpression of the eight cell surface antigens, which serve as binding platforms for the nine currently approved ADCs, another important element of these antigens is that they are intended to be entry portals into the intracellular space, which is required for drug release and activity. In fact, ADC efficacy is dependent on the efficiency of target-mediated internalization to deliver the payload inside tumor cells [6,7]. Moreover, ADCs are required to deliver their cytotoxic payloads inside tumor cells at concentrations sufficient to evoke effective cytotoxicity [8]. ADCs must (1) bind to the target cell surface receptor; (2) efficiently internalize into the cytoplasmic portion of the cell; (3) be transported through the endosomal-lysosomal pathway with final delivery into functional lysosomes; and (4) be completely catabolized for the release of active payload metabolites that can either permeate or are likely transported out of lysosomes via a lysosomal membrane transporter to interact with their molecular targets in the cytoplasm or nucleus [9,10,11,12,13]. The early ADCs such as T-DM1 and GO were developed based on antibodies that were selected based on receptor–receptor inhibition or for delivering radionuclide therapy from the surface of the cell, respectively, and were not necessarily focused on internalization.

This review is motivated by the surprisingly limited information concerning the mechanistic endocytic processes of target receptors, including when bound to antibodies or ADCs. Numerous reviews exist that provide excellent general overviews of ADCs for cancer therapy [9,10,14,15,16]. In addition, reviews exist that provide more focused descriptions of linker and conjugation chemistries [17,18,19,20], current and emerging payloads [10,21], and ADC resistance mechanisms [10,22,23]. With respect to the actions of ADCs once internalized, there are enlightening reviews that describe the intracellular trafficking dynamics and their implications for ADC efficacy [6,24,25]. Intracellular trafficking dynamics is an area of intense investigation as there is debate on whether cleavable, pH/reduction-sensitive releasable, or non-cleavable linkers are preferred. This is further complicated by the question of whether the payload (or its active metabolite) is able to diffuse across membranes to provide a ‘bystander effect’, and whether this improves tumor killing or contributes to dose-limiting toxicities, and the cancer type (hematological or solid). Chalouni and Doll recently described molecular markers and mechanisms for endocytosis and the intracellular sorting of ADCs [24]. Moreover, they listed several related markers and their role in cancer. However, these endocytosis and intracellular sorting markers were not specifically associated with the endocytic activities of the approved ADCs or their target receptors, most likely because there is limited information available on this topic.

It is essential to understand endocytosis, as it is the first stage of the ADC intracellular delivery process. This review does not attempt to provide a comprehensive description of normal endocytosis pathways, which can be found in [26,27]. In the past decade, the concept of derailed endocytosis has recently emerged as a hallmark of cancer cells, whereby examples have emerged of aberrant internalization and endosomal trafficking events that drive cancer initiation and progression [28,29]. Many discoveries on endocytosis have occurred due to experiments using mAbs as tools to progress endocytosis research. When there is perturbation of endocytosis that affects the internalization of a receptor, it is reasoned that the perturbation will accordingly affect ADC internalization. It is increasingly likely that distinct adaptor and accessory proteins involved in the major endocytic pathways control the internalization of distinct cell surface receptors [26]. Therefore, it is also highly likely that unique adaptor and accessory proteins may play a role in inhibiting ADC internalization when cancer cells acquire resistance.

Therefore, this review seeks to consolidate information and provide insight into important aspects of endocytosis, providing (1) a description of the endocytic pathways (or lack thereof) utilized by the receptors in normal and/or tumor cells targeted by the nine currently approved ADCs; (2) a description of unique endocytosis features when receptors are bound by ADCs (including investigational ADCs and antibody-based agents); (3) an overview of the fragmentary information available concerning these endocytosis pathways as ADC-specific resistance mechanisms; and (4) our perspective on endocytosis and its clinical implications. Target-independent uptake is a significant occurrence due to the Fc portion of mAbs and further information on its implications for ADC efficacy and toxicity is reviewed in [30]. The present review focuses strictly on target-dependent ADC endocytosis.

## 2. Relevant Endocytic Pathways for Currently Approved ADCs

In general, normal endocytosis can be categorized into three stages, (1) bud formation, (2) membrane curvature and vesicle maturation, and (3) scission and release of the membrane into the cytoplasm. There are multiple endocytosis pathways that have overlapping aspects, and hence the general process of endocytosis is highly flexible and complex. As a result, current ADC research is only scratching the surface of the relationship between endocytosis and ADC efficacy. Here, we briefly describe the implicated endocytosis pathways and their components that are pertinent to our findings regarding endocytosis in relation to the nine approved ADCs.

### 2.1. Clathrin-Mediated Endocytosis

Clathrin-mediated endocytosis (CME) is conceptually a straightforward process that includes a few sequential and partially overlapping steps. CME can occur constitutively for certain receptors at the plasma membrane, whereas other receptors require ligand and/or antibody binding to initiate CME or enhance its activity. CME commences when endocytic coat proteins from the cytoplasm begin to cluster on the inner leaflet of the plasma membrane. The protein coat continues to assemble and grow by recruiting and interacting with additional protein adaptors from the cytoplasm. Key adaptor proteins enable membrane bending and hence concentrate the internalizing receptors/ligands into a ‘clathrin-coated pit’ (CCP). Separation from the plasma membrane occurs through a scission process at the point where the CCP neck becomes constricted due to the CCP invagination growing larger. Actin polymerization contributes to pulling the CCP ‘inwards’ into the cytoplasm until scission is complete and the CCP is released and becomes a clathrin-coated vesicle (CCV). Finally, the CCV coat disassembles and the CCV fuses with endosomes for sorting to defined subcellular locations, or can be recycled back to the cell surface.

Clathrin is the key component of CME and is composed of a heavy and light chain. Three clathrin heavy and light chains form a trimeric clathrin ‘triskelion’ that interacts with other triskelions and forms a polygonal lattice around the emerging CCP. More information on clathrin’s structure and function is reviewed in [27]. There are more than 50 additional cytosolic proteins that are involved in the formation of CCVs [27]. These proteins assemble from the cytoplasm to the site of endocytosis in a highly ordered manner. The adaptor protein 2 (AP-2) is a heterotetrameric complex and is a major adaptor as it binds to lipids in the plasma membrane and to cargo molecules [27]. Adaptor proteins, such as the endophilins (described in Section 4.1), that contain Bin/Amphiphysin/Rvs (BAR) domains mediate the constriction of the CCP neck. The BAR domain-containing proteins interact with dynamin to mediate scission. Dynamin is a GTPase, which forms a helical polymer around the neck of the mature vesicle. Upon GTP hydrolysis, dynamin induces fission of the vesicle from the plasma membrane [31].

### 2.2. Clathrin-Independent Endocytosis

Clathrin-independent endocytosis mediates the cellular uptake of several extracellular ligands, receptors, pathogens, and as we will describe, potentially ADCs. The understanding of clathrin-independent endocytic systems is not nearly as comprehensive as our understanding of CME [26]. Furthermore, in the context of ADC internalization, even less is known. We provide a brief overview of three systems we found that may be relevant for ADC internalization, namely, caveolae-mediated endocytosis, caveolin-independent carriers/GPI-anchored protein-enriched early endosomal compartments (CLIC/GEEC), and macropinocytosis.

#### 2.2.1. Caveolae-Mediated Endocytosis

Caveolae are small flask-shaped invaginations of the plasma membrane, characterized by high levels of cholesterol and glycosphingolipids that mediate endocytosis via clathrin-independent pathways and are found in most cell types [26]. The principal scaffolding proteins of caveolae are the caveolins, which are 20–24 kDa integral membrane proteins that form oligo multimers [26]. Caveolin-1 is the best characterized due to its presence in a variety of cell types, whereas caveolins-2 and -3 are expressed primarily in adipocytes and striated muscle cells, respectively [32,33,34]. The caveolins share common scaffolding domains that mediate interactions with themselves and other proteins that contain caveolin-binding domains [35]. Normally, caveolin-1 forms complexes of 14–16 monomers at caveolae invaginations [36]. Caveolin-1 also has increased levels of phosphorylation, which is required to induce caveolae-mediated endocytosis [37]. The membrane curvature required to form the bulbs is governed by coordinated and cooperative interactions between the caveolins, and additional core proteins such as cavins and pacsins/syndapins. Dynamin recruitment to the neck region of caveolae, which is needed for scission, has been demonstrated [38]. Caveolae are enriched with cholesterol and specific glycosphingolipids and are thought to be specialized lipid domains.

Although caveolae share an invagination-like morphology that is similar to that of CCPs, they are distinct and are reviewed in [39]. Briefly, CCP density is constant, whereas caveolae density can vary greatly depending on cell type. CCPs grow bigger as the budding endosome matures. In contrast, caveolae bulbs maintain a constant size. Once inside the cell, caveolae form higher-order structures and not the simple spherical shapes of endosomes formed from CCPs. Additional unique aspects of caveolin-mediated endocytosis are that only ~1% of caveolae are estimated to bud from the plasma membrane [40]. Of the small percentage of internalized caveolae, it appears to follow a recycling pathway where it colocalizes with Rab5 (a marker of early endosomes) [41]. This may pose challenges for ADCs that target receptors that utilize caveolin-mediated endocytosis. Lastly, caveolae components and their overexpression have been linked to cancer [42,43], but there is limited information on how this affects the rate of internalization.

#### 2.2.2. CLIC/GEEC Endocytosis

CLIC/GEEC is an endocytic compartment that occurs mainly in ligand-activated cells, which can be caused by growth factors, receptor crosslinking by antibodies, or by bacterial toxins and viruses [44]. In addition, cellular membranes must be in a state of high fluidity as CLIC/GEEC does not function below physiological temperatures or when the membrane is under increased tension. CLIC is increased at the leading edge of migrating cells [45]. Other relevant parameters that identify CLIC/GEEC pathways include dynamin-independent plasma membrane scission [46], sensitivity to cholesterol depletion [40,47], the acquisition of Rab5/merging with early endosomes, placental alkaline phosphatase (PLAP), and the GTPase regulator associated with FAK (GRAF1) [48,49]. For CLIC, GRAF1 controls their formation as it contains BAR domains that induce membrane curvature and facilitate plasma membrane invagination and scission [44].

#### 2.2.3. Macropinocytosis

Macropinocytosis describes a form of larger-scale endocytosis that frequently involves highly ruffled regions/protrusions of the plasma membrane that subsequently fuse with one another or back with the plasma membrane [26]. Membrane ruffling is the morphological hallmark of macropinocytosis. This process results in the uptake of extracellular fluid and the components trapped within these sites. Macropinocytosis is dependent on actin polymerization and the proteins Rac1 and p21-activated kinase 1 (PAK1). PAK1 is a key regulator as it interacts with Rac1, which activates phosphatidylinositol-3-kinase (PI3K), Ras, Src, and Hsp90 to promote macropinocytosis [50,51,52,53,54]. Macropinocytosis is also cholesterol-dependent, which is required for the recruitment of Rac1 [55]. These components eventually result in a process that involves the uptake of larger membrane areas than those of CME and caveolae mechanisms.

## 3. Target Antigens for Approved ADCs and Their Endocytosis Characteristics

In the following sections we describe the current understanding of the endocytosis activities of the targets for the currently approved ADCs. In addition, we summarize what has been demonstrated in relation to mAb/ADC binding and endocytosis and intracellular trafficking.

### 3.1. CD33

CD33 is a 67-kDa transmembrane glycoprotein receptor that is commonly expressed on normal myeloid cells and is the target for GO due to its preferential overexpression on AML cells. CD33 is a member of the sialic acid-binding immunoglobulin-like lectin (Siglec) family and is also known as Siglec-3. The characteristic structural feature of Siglec cells is their extracellular *N*-terminal V-set immunoglobulin domain (Figure 1), which recognizes and binds glycans. This is followed by one C2-type immunoglobulin domain, a transmembrane helical sequence connected to a cytoplasmic domain containing two immunoreceptor tyrosine-based inhibitor motifs (ITIM) [56]. The primary role of the ITIM motifs is to modulate signaling in order to downregulate leukocyte activation [57]. Briefly, upon glycan binding to CD33, the phosphorylated tyrosines (positions 340 and 358) transmit inhibitory signals by binding and activating Src homology-2 (SH2) domain-containing tyrosine phosphatases (SHP-1 and SHP-2), which dephosphorylate intracellular substrates that have a role in cell activation [58,59,60].

However, the ITIMs also regulate CD33 endocytosis. ITIM binding by SHP activates endocytosis via CME but in the absence of AP-2 (Figure 1) [59,61,62]. Endocytosis was increased when CD33 was phosphorylated, relative to non-phosphorylated cells, in the AML cell lines NB4, HL-60, ML-1, U937, and TF-1 [62]. Phenylalanine substitutions for the ITIM tyrosines blocked binding to SHP-1 and SHP-2 and significantly reduced endocytosis [58,61]. In cells containing CD33 with non-manipulated phosphorylation states, SHP-1 and SHP-2 are dominant in dephosphorylating CD33.

For internalization efficiency, there is no correlation between the expression level of CD33 and its internalization rate in AML cells [61]. CD33 is classified as a slow internalizing antigen. This is based on the results of internalization assays that revealed that approximately 50% of CD33 molecules were internalized over 2 h in Jurkat cells, which was significantly slower when compared to internalization rates for the low-density lipoprotein receptor, which shows natural rapid endocytosis. Furthermore, CD33 crosslinking did not improve endocytosis [61]. Taken together, ITIM phosphorylation is required for endocytosis but there is a dominating presence of tyrosine phosphatases that dephosphorylate CD33. As a result, CD33 has a slow internalization rate.

Walter et al. surmised that a potential cause of the slowness of CD33 endocytosis was its low abundance of tyrosine-based motifs [61]. For example, the related Siglec CD22 (described in Section 3.5) contains multiple ITIM and ITIM-like motifs [63]. Unlike CD33, CD22 undergoes constitutive endocytosis (described in Section 3.5). It is possible that CD33 does not contain sufficient ITIMs that would provide a balance that favors the phosphorylated (internalizing) state. Hence, increased phosphorylation of Siglec ITIMs favors the recruitment of the CME machinery for endocytosis but is outcompeted by phosphatases required for signaling.

### 3.2. ADC/Ab-CD33 Endocytosis

Impaired CD33 internalization is strongly suggested as a significant reason underlying the ineffectiveness of GO. AML cells taken from patients administered with GO showed only 25–40% internalization of CD33 3–6 h post-injection [64]. This internalization rate based on in vivo treatment was comparable to the in vitro findings previously described on AML cell lines [61]. However, it was surmised that CD33 underwent significant renewal after endocytosis. Indeed, it was determined that ≥25% of CD33 underwent renewed expression within the 6-h GO treatment window. GO was observed in the cytoplasm after only 30 min and by 24 h it disappeared, which indicated that a fraction of GO must be rapidly internalized and subsequently catabolized in lysosomes. Additional anti-CD33 mAbs, such as M195, have shown that CD33 cell surface levels were also significantly renewed after treatment [65]. Jedema et al. reported that CD33 endocytosis and renewal can be increased if AML cells are activated by treatment with GM-CSF or IFN-γ [66]. These studies indicate that although CD33 may undergo slow internalization, CD33 renewal enables a sufficient fraction of GO to be internalized and catabolized, which is required for it to kill AML cells effectively.

Interestingly, Paubelle et al. demonstrated that patients with mutations H63D or C282Y in a gene called HFE (most common cause of hereditary hemochromatosis) had poor overall survival rates compared to patients with wild-type HFE [67]. Their studies showed that in AML cells taken from patients with mutated HFE, there was significantly less CD33 internalization, which resulted in reduced GO cytotoxicity. It was also shown that the suppressor of cytokine signaling 3 interacts with internalized and phosphorylated CD33, leading to transport to and degradation in the proteasome [68]. It was suggested that this mechanism sequesters pools of phosphorylated CD33 (most likely to internalize it) and is most likely a mechanism for blocking anti-CD33 therapy.

CD33 clearly undergoes endocytosis and, hence, GO is also internalized and routed to lysosomes, where it can successfully release its payload and kill AML cells. Nonetheless, overall it appears that CD33 undergoes slow internalization or degradation, which is further exacerbated by the presence of CD33-associated proteins that harbor mutations relevant to poor internalization and most likely contributes to the controversial effects of GO (Table 2).

### 3.3. CD30

CD30 is a 120-kDa transmembrane glycoprotein and belongs to the tumor necrosis factor receptor (TNFR) superfamily. Its extracellular portion is comprised of six cysteine-rich domains (CRD) in an extended conformation (Figure 1) [69]. The membrane proximal CRD is known to mediate the ligand-independent assembly of CD30 oligomers through its preligand assembly domain and ligand binding induces the formation of higher-order receptor clusters [70]. These complexes are necessary for efficient signaling [71,72,73]. The role of clustering in the context of endocytosis is unknown.

CD30 is expressed on activated T and B cells and on various lymphoid neoplasms, including Hodgkin lymphoma and ALCL [74]. The CD30 ligand (CD30L) is expressed on various lymphoid cells. CD30–CD30L binding induces various biological effects on CD30-positive cells, including activation, proliferation, differentiation, and cell death in lymphoma cell lines [75]. More about CD30 and its role in health and disease can be found in [74].

CD30 is not known for endocytosis and, in contrast, is well known for shedding due to proteolytic cleavage, and its release is enhanced upon interaction with CD30L-positive cells. CD30 shedding is mediated by matrix metalloproteinases (MMPs), including TNF-α converting enzyme (also known as ADAM17) and ADAM10 in lymphoma cells [76,77]. Lymphoma cell lines and normal peripheral blood leukocytes in media underwent constitutive CD30 shedding when artificially stimulated with phorbol 12-myristate 13-acetate, which stimulates ligand binding. Shedding was blocked in the presence of MMP inhibitors. Shedding was reliant on the presence of CDR 2 and 5, which have nearly identical sequences [78]. Interestingly, shedding is enhanced by most anti-CD30 mAbs that target CRD 1 and 6 (mAbs Ki-1 and Ki-2), but is reduced in mAbs that target CRD 2 and 5 (mAbs Ki-4 and Ber-H2). Because MMPs cleave CD30 at the juxtamembrane stalk, mAbs that bind domains 2 and 5, which are some distance away from the cleavage site, reduce but do not eliminate shedding. Lastly, shed CD30 is able to induce more shedding by directly interacting with membrane-bound CD30 [79].

Inhibiting CD30 shedding has been investigated to improve the efficacy of CD30-targeted therapeutics. The hydroxamic acid-based inhibitor BB-3644 reduced soluble CD30 levels in the blood in preclinical studies [80]. However, BB-3644 inhibits additional MMPs and contributed to dose-limiting musculoskeletal toxicities in human trials [81]. More selective ADAM10 inhibitors are in development, with promising improvements in off-target toxicities [82].

Shedding is such a feature of CD30 biology that elevated concentrations of circulating soluble CD30 are sufficiently abundant that they can be used as a serum marker for monitoring tumor progression [78]. CD30 may require shedding to maintain normal human health. For example, in certain autoinflammatory syndromes, the CD30 family member TNFR1 has a single mutation that blocks MMP cleavage and results in an elevated receptor density [83]. For ADC efficacy, elevated CD30 circulating levels would appear to sequester injected ADC and hence reduce the amount of ADC that is able to localize to CD30-positive tumor sites. Hence, the lack of endocytosis findings based on natural activities related to ligand binding would suggest that CD30 is not an ideal ADC target.

### 3.4. ADC/Ab-CD30 Endocytosis

There is a disconnect between CD30 endocytosis/Ab internalization and therapeutic efficacy. Previous reviews focusing on CD30 as a therapeutic target have claimed that CD30 undergoes rapid endocytosis [74,84]. However, these claims are based solely on a report by Sutherland et al. which reported on mAb-induced CD30 internalization [85]. The CD30-specific mAbs Ki-1, Ki-2, Ki-3, Ki-4, Ber-H2, 5F11 (also known as MDX-060 and iratumumab), and AC10 have all been developed as therapeutics for lymphoma. The latter mAb was developed into BV. Our readings do not find clear evidence for how CD30 targeting by ADCs results in cytotoxicity.

An additional complication is that there are several anti-CD30 mAbs reported in the literature that bind different domains. The mAbs Ki-4, Ber-H2, and 5F11 were first shown to be able to bind to CD30 CRD 2 and 5 and reduce CD30 shedding [78,79]. In contrast, mAb Ki-1 and Ki-2 bound to CRD 1 and 6, enhancing CD30 shedding. Although 5F11 showed promising preclinical therapeutic results, it failed in the clinic due to lack of efficacy (comprehensively detailed in [74]). Ki-3, Ki-4, and Ber-H2 were developed as immunotoxins but also suffered from poor efficacy. As immunotoxins also need to effectively enter target cells, one reason for this lack of efficacy was that the Ki-3 conjugate only internalized 3.9% of the total loaded conjugate [80]. Hence, very limited information on CD30 endocytosis in context with these mAbs has been reported, and CD30 shedding likely explains why these unmodified mAbs or conjugates lacked clinical efficacy.

As previously mentioned, Sutherland et al. did show that humanized AC10 conjugated to monomethyl auristatin A and F (MMAE and MMAF, respectively) during the early preclinical development of what became BV, and did show internalization, colocalization with lysosomes, and highly cytotoxic effects [85]. In addition, CD30 was identified to undergo CME (Figure 1) as internalization was blocked when cells were treated with multiple types of CME inhibitors. Caveolae-mediated inhibitors had no effect on endocytosis. However, the endocytosis of CD30 is unclear, primarily, due to the method of the internalization assay. AC10 ADC versions were incubated with CD30-positive L540 lymphoma cells for 30 min on ice and then rinsed. The cells were then incubated with goat anti-human IgG to induce crosslinking. Cells were then washed and shifted to 37 °C. Cells were then prepped for fluorescence microscopy with fluorescently labeled antibodies to detect AC10. This assay produced confusing information in two ways. First, internalization was aided through crosslinking as a theorized means of promoting CD30 clustering and, subsequently, internalization. Second, the reduction of cell surface CD30 was most likely attributable to shedding and not to internalization. Nonetheless, Okeley et al. showed that BV did release MMAE inside lymphoma cells and quantified the amount of intracellular drug [86]. Furthermore, it was shown that the number of ADCs internalized and catabolized was 2.5 and 3.4 ADCs per CD30 receptor for Karpas299 and L540cy cells, respectively. Importantly, the ADC catabolism increased over time. This suggested that the lymphoma cells evaluated were able to renew CD30 in order to maintain the increased ADC catabolism post-24 h.

Structural data on mAb binding to distinct CRDs would be valuable in order to determine the unique interactions that could explain why certain mAbs–CRD interactions promote CD30 shedding and to clarify which interactions could promote endocytosis, supposedly via receptor clustering.

### 3.5. CD22

CD22 is a 140-kDa transmembrane glycoprotein, and like CD33 it is also a member of the Siglec family and shares multiple structural features with this family, specifically extracellular Ig domains capped by a V-set Ig ligand binding domain, and an intracellular region containing ITIM motifs [87]. Key differences are that CD22 is much larger than CD33 due to its multiple Ig domains and ITIM/ITIM-like motifs (Figure 1). CD22 expression is restricted to B cells. The ITIMs recruit phosphatases such as SHP-1, which inhibits B-cell receptor (BCR)-induced signaling in order to regulate B-cell homeostasis [88]. CD22 is expressed at increased levels in the majority of blast cells of various B-cell malignancies, including ALL [87].

CD22 undergoes endocytosis via CME (Figure 1) [89,90,91]. The ITIMs of CD22 interact with the AP50 subunit of AP-2, and tyrosine-843 and -863 are required for interaction and subsequent endocytosis [89,91]. Residues arginine-737 and glutamine-739 are also critical for internalization, and alanine mutations of these residues abrogate endocytosis [89]. CD22 does not utilize caveolae-mediated endocytosis as there is no disruption in internalization rates in the presence of the caveolin-inhibitor nystatin.

The Paulson group published a collection of papers that elegantly evaluated CD22 endocytosis in a more natural setting [89,90,92]. The group developed a decavalent scaffold composed of the preferred glycan sequence (NeuAcα2-6Galβ1-4GlcNAc), which binds to CD22 and induces receptor clustering. Using this system, the ligands accumulated inside the cells at 37 °C at an order-of-magnitude greater level compared to cells at 4 °C. Ligand intracellular accumulation was blocked in the presence of bafilomycin A, a proton pump inhibitor, indicating that the ligands were destined for lysosomal degradation. Interestingly, there was only a 2-fold intracellular increase in CD22 in cells incubated at 37 °C relative to those at 4 °C. Moreover, the amount of intracellular CD22 never surpassed 50% of the total CD22 surface levels at all tested time points. The group determined that B cells contained a large intracellular storage of CD22, which localizes to the cell surface and compensates for internalized CD22. Lastly, the rate of CD22 internalization was not enhanced by these ligands, indicating constitutive endocytosis.

Taken together, natural-like ligands accumulate within cells via the constitutive and rapid internalization of CD22. These ligands are sorted for degradation in lysosomes, while CD22 recycles back to the cell surface. In addition, CD22 ligand-induced endocytosis activates intracellular pools, which replenish or increase the level of CD22 expression on the cell surface. Hence, CD22 has excellent endocytosis properties for ADCs.

### 3.6. ADC/Ab-CD22 Endocytosis

The rapid internalization of anti-CD22 mAbs on a variety of B cell lines has been known for the past two decades [93,94], yet it remains controversial whether or not bound antibody-based agents are sorted in a similar fashion as natural glycan ligands. Time course incubation of the mAb RFB4 with CD22-positive sultan cells showed an internalization half-life of <1 h [93]. The mAbs colocalized with early endosomes at time points ≤30 min, but progressively accumulated in late endosomes and lysosomes past 30 min. This indicated that bound mAbs/Ab-based agents undergo the same intracellular routing as bound glycan ligands, and this delivery system is optimal for the development of ADCs.

The mAbs RFB4 and epratuzumab have been previously studied in the context of endocytosis and intracellular routing. O’Reilly et al. showed that, unlike the ligand NeuAcα2-6Galβ1-4GlcNAc, RFB4:CD22 did not separate after endocytosis and recycled back to the cell surface [92]. Erena-Orbea et al. reported on the crystal structure of CD22 bound to epratuzumab Fab [88]. It was shown that epratuzumab bound at the interface of domains 2 and 3. The orientation led the authors to conclude that epratuzumab most likely promotes CD22 clustering via antibody crosslinking and promotes internalization. However, experiments testing binding at various pH values showed that epratuzumab remained bound to CD22 at pH values of 5.5–6.5 and 4.5–5.5, which mimicked early and late endosomes/lysosomes, respectively. These findings support the notion that unlike glycan ligands, bound mAbs are recycled and not sorted to lysosomes.

Du et al. showed a direct relationship between endocytosis and cytotoxicity [95]. RFB4 was developed as an anti-CD22 immunotoxin where the single-chain Fv was genetically fused to a *Pseudomonas* exotoxin. It was shown that when 100 nmol/L of the RFB4 immunotoxin was incubated with CD22-positive cells, there was an increase of up to 240% in the level of immunotoxin that accumulated inside the cells. This indicated that the amount of CD22 internalized was greater than the original cell surface levels by 2–3-fold. Quantification of the intracellular CD22 revealed levels exceeding cell surface levels by up to 140%. Shan et al. showed that the mAb HD39 did colocalize with early endosomes at early time points [93]. However, at later time points the mAb was sorted and colocalized with late endosomes/lysosomes.

G544, the mAb portion of InO, targets epitope A, which is located at the *N*-terminal domain and is also thought to promote crosslinking [87]. Fingrut et al., recently presented a case of a patient with low CD22 expression on leukemic blasts (<30% of normal levels) that showed a very good response and survival benefit from InO treatment [96]. Based on the impressive amount of work and the observation that InO is obviously effective, even if InO is recycled back to the cell surface, the outstanding internalization activity of CD22 compensates for the limited amount of InO that is sorted to late endosomes/lysosomes. It has been proposed that the epitope to which it is bound can influence antibody sorting, and G544 may bind an epitope that sorts it to lysosomes instead of the cell surface, as occurs with epratuzumab. A detailed analysis of the crystal structure of G544:CD22 will provide further understanding on this matter. Nonetheless, our findings suggest that CD22 and bound ADCs will be efficiently internalized.

### 3.7. CD79b

BCR endocytosis is the gateway to targeting BCR-bound antigens to lysosomes as part of the antigen processing and presentation pathway of B cells to recruit T cell help. DLBCL is the most common subtype of non-Hodgkin lymphomas, constituting up to 40% of cases globally [97]. CD79b is exclusively expressed in immature and mature B cells and is overexpressed in ≥80% of neoplasms [98,99].

Specific antigen binding occurs via the cell surface immunoglobulin portion of the BCR. However, CD79a and CD79b, two non-covalently associated transmembrane proteins, mediate signaling and endocytosis. For the latter function, the CD79a-CD79b heterodimer is a scaffold that controls BCR endocytosis. The combined intracellular regions of CD79a and CD79b possess five immunoreceptor tyrosine-based activation motif (ITAM)-signaling modules [100]. Interestingly, ITAM phosphorylation, and hence signaling, occurs in only a small fraction of antigen-bound BCR molecules, which are then retained at the cell surface. The large fraction of the unphosphorylated antigen-BCR complex is dedicated for endocytosis and trafficking to the major histocompatibility complex class II-positive compartment, a lysosome-like vesicle [101,102].

BCR endocytosis is primarily accomplished by CME and is mediated by AP-2 [103,104,105,106]. Interestingly, it is CD79a that directly interacts with the μ subunit of AP-2, which in turn activates CD79b and leads to the internalization of the entire BCR complex [100]. In addition, it is perhaps consequential for ADCs that although CD79a is able to be internalized as a monomer, CD79b is not [107,108,109,110]. If the proximal membrane tyrosine (Y195) of CD79b is mutated, AP-2 binding by CD79a is blocked, as is endocytosis. In 18% of activated B-cell-like DLBCL specimens, Y195 was mutated [111]. Activated BCR was also shown to localize with lipid rafts as part of its internalization process [112].

Taken together, CD79b shows the appropriate expression and efficient internalization and trafficking to lysosome-like compartments, but evidence suggests that its endocytic activity is dependent on the internalization of the entire BCR complex, rather than its internalization as a monomer. Furthermore, one can posit that an anti-CD79a ADC would be just as effective as other ADC targets due to its ability to internalize as a monomer.

### 3.8. ADC/Ab-CD79b Endocytosis

Anti-CD79b mAbs, specifically clone SN8, have been shown to bind and to be efficiently internalized into lymphoma cells for nearly three decades [113]. SN8 was eventually transformed into PV. Early testing of SN8 as an ADC evaluated its performance when conjugated to the microtubule inhibitors DM1 or MMAF via stable linkers [114]. SN8 was modified via lysines using SMCC-DM1 or cysteines to MMAF using a non-cleavable maleimidocaproyl linker. Both ADC formats had similar cytotoxic and anti-tumor potencies in multiple NHL cell lines/xenografts. There was no correlation with CD79b expression and tumor killing. A key finding was that the anti-CD79b ADC was significantly more potent than an anti-CD79a ADC that was also developed and tested. The decision on whether to pursue CD79a or b was based on tumor-bearing mice administered with single intravenous injections of each ADC at a dose of 64 μg/kg. Only the anti-CD79b ADCs were found to regress tumor volumes. In contrast, the anti-CD79a ADCs were able to prevent tumor growth for a short period, after which tumors grew again. In addition, the SN8 ADC was rapidly internalized and showed strong accumulation in the major histocompatibility complex class II-positive compartment in vivo and in vitro. Mice injected with SN8-ADC had their tumors excised and single-cell suspensions prepared for flow cytometric analysis. It was found that the surface expression of the BCR via probing of CD79a and IgM were substantially reduced in the tumors treated with the anti-CD79b ADCs. In vitro, the treatment of several lymphoma cell lines showed that 100% SN8 mAbs were rapidly internalized within 20 min of incubation at 37 °C. Furthermore, SN8 colocalized with LAMP1 1 h after uptake and accumulated in the lysosomal-like compartments within 3 h. Importantly, there was no colocalization with transferrin, which is a known marker of recycling endosomes. In comparison, ADCs conjugated to their payloads with a non-cleavable linker against targets that had poor internalization were not effective [115]. This study was significant, because it did support that CD79b endocytosis is linked to the simultaneous internalization of the entire BCR complex and that its internalization does not occur as a monomer. The study also provided the discovery that CD79b, and not CD79a, is the more important target for an ADC drug. Lastly, the authors suggested that because CD79b rapidly internalizes and appears to be completely trafficked to lysosomal compartments, a stable linker that requires complete protease digestion is the ideal ADC format.

Although efficient CD79b endocytosis and lysosomal trafficking is important, it cannot compensate for expression below a minimum threshold. SN8 was humanized and engineered using the THIOMAB system and, in contrast to the previously described method, was then conjugated to MMAE via the cathepsin-cleavable vc-PAB linker [116]. Dornan et al. reported that below a threshold <6.82 geometric mean fluorescence intensity units, CD79b-positive lymphoma cell lines were insensitive to anti-CD79b ADCs at high concentrations of 10 μg/mL [116]. Evaluating primary samples from patients with chronic lymphocytic leukemia, marginal zone lymphoma, hairy cell leukemia, follicular lymphoma, mantle cell lymphoma, and DLBCL showed that CD79b was expressed above the threshold in almost all cases. In particular, 90% of DLBLC specimens were above the threshold for CD79b expression, and this supports the previously-described earlier findings on CD79b expression in DLBCL cells. Again, impressive ADC internalization was observed. Nearly 100% of the anti-CD79b-vc-PAB-MMAE ADC was internalized in the first hour in DLBCL cell lines that represented the two major subtypes—the germinal center and activated B-cell-like DLBCL [116].

CD79b mutations occur in up to 25% of activated B-cell subtypes of DLBCL and have been thought to inhibit endocytosis, particularly when mutations occur in the ITAM region [117,118]. The anti-CD79b-vc-PAB-MMAE ADC was also able to internalize and kill the TMD8 cell line, which harbors the Y196H mutation, as efficiently in cell lines with wild-type CD79b. Notably, this mutation was not the essential mutation at position Y195 that was shown to block CD79b endocytosis [100]. Moreover, the mutations in CD79B almost exclusively affect Y195 [111].

In summary, CD79b appears to be tailor-made for the ADC mechanism of action. Its exquisite endocytosis and subsequent trafficking to protease-rich lysosome-like compartments would be an ideal means for ADCs to efficiently deliver their payloads and to cause effective cytotoxicity. Given that the expression of CD79b is highly expressed in most DLBCL specimens, this signifies that most patients will be most likely to respond to PV. The current primary reason for a potential absence of response appears to be low expression, and although it has been shown that there is no correlation between CD79b expression and response, patients with tumor cells above a certain threshold will most likely respond to PV therapy.

### 3.9. Trop2

Trop2 is a 46-kDa monomeric glycoprotein that has properties such as preferential overexpression, constitutive endocytosis, and routing to lysosomes that make it a highly attractive target for ADCs. A review of Trop2 and its function in normal and diseased cells can be found in [119]. Trop2 overexpression on the cell surface is well documented, and it has been identified in approximately 30 different tumor types [119]. However, Trop2 is interesting due to the amount of protein detected in the cytoplasm of tumor cells and the association with patient prognosis. Considerable Trop2 expression in the cytoplasm is detected in nearly every tumor type, where expression is noted at the cell surface. However, what is the significance of such a cytoplasmic presence with endocytosis and, furthermore, how does it relate to the therapeutic effectiveness of ADCs?

Trerotola et al. first observed that intracellular Trop2 was specifically localized in the endoplasmic reticulum and the Golgi apparatus in normal and tumor tissues (including tumor cell lines) [120]. However, an abundance of the Trop2 protein was also detected in early and late endosomes. One important feature is that the skin contained the highest overall Trop2 staining intensity (immunohistochemistry [IHC] score ++/+++). Strop et al. later determined normal keratinocytes to have 50,000 Trop2 copies per cell. In contrast, cell lines from various different tumor types contained 2–10-fold more Trop2 copies per cell [121]. Most likely, the amount of intracellular Trop2 is also accordingly increased in tumor cells relative to normal cells.

The internalization mechanism for Trop2 has been linked to CME (Figure 1), which promotes cellular proliferation. Wanger et al. demonstrated that treatment of HEK293 cells with dynasore, the small molecule inhibitor of dynamin, blocked Trop2 endocytosis [122]. Working in a prostate cancer model, it was observed that Trop2 undergoes continual rapid endocytosis. Trop2 was sorted to both early and late endosomes. Interestingly, Trop2 was cleaved in a portion of the endosomes. These endosomes then trafficked to the cell surface, where they budded and where secreted in the form of exosomes into the media. Importantly, the Trop2 cleavage products have been shown to have oncogenic signaling potential, as they bind β-catenin, which upregulates cyclin D1 and promotes c-myc expression [123]. The Trop2 cleavage products were found in aggressive PC3 prostate cancer cells but not in LNCaP cells, indicating that the rapid internalization process could occur in aggressive prostate cancers. SG is currently under investigation in a single-arm, open-label, multicenter phase 2 trial in patients with castrate-resistant prostate cancer (NCT03725761).

A potential explanation for the robust endocytosis observed with Trop2 could be due to significant Trop2 clustering. Fu et al. showed that Trop2 was almost exclusively clustered on the surface of A549 adenocarcinoma cells and the clustering occurred more on the apical membrane than the basal membrane [124]. Moreover, clustering was also prominent on human bronchial epithelial cells but was approximately 2-fold lower compared to A549 cells. Depolymerization of the actin cytoskeleton with cytochalasin B and the disruption of lipid rafts with methyl-β-cyclodextrin to remove cholesterol dramatically decreased the amount of Trop2 clusters. Actin and lipid rafts are known to be actively involved in dynamin-mediated endocytosis [125].

Trop2′s conformational dynamics has been studied, and it has been shown that Trop2 forms a natural homodimer through an interaction segment composed of the amino acids ‘VVVVV’ located in the transmembrane domain [126]. Pavic et al. further suggested that dimerization of Trop2 can further recruit Trop2 monomers into close proximity via other cell surface proteins [126]. Thus, Trop2 clusters are most likely formed by multiple dimers linked via lipid rafts and other membrane-bound proteins.

Trop2 binds several ligands, such as claudin-1, claudin-7, cyclin D1, and insulin-like growth factor 1 (IGF1), and neuregulin-1 (NRG) cells have been reported to bind and activate Trop2 and subsequent downstream signaling pathways [127,128,129]. However, none of these ligands have demonstrated that Trop2 is internalized upon binding or interaction. Zhang et al. suggested that NRG interaction with Trop2 induces endocytosis as part of a downregulation process in order for NRG to be cleaved and released from the cell, where it can then bind to EGFR3 and activate head and neck squamous cell carcinoma progression [129].

Thus, Trop2 undergoes endocytosis that is more robust in tumor cells compared to normal cells. In combination with the preferential overexpression of Trop2 on tumor cells, this suggests that Trop2 is an excellent target for ADCs.

### 3.10. ADC/Ab-Trop2 Endocytosis

The mAb ‘RS7′, which would eventually be humanized and become the targeting component of SG, was originally produced from a hybridoma derived from a surgically removed human primary squamous cell carcinoma of the lung and was found to be reactive to cell lines representing various tumor types [130]. Shih et al. and Stein et al. later demonstrated that radiolabeled RS7 could bind and be internalized in the cell lines MDA-MB-468 and Calu-3 that represented breast carcinoma and lung adenocarcinoma, respectively [131,132]. Internalization was determined by fluorescence microscopy and compared to an anti-transferrin mAb, the classic example of a rapidly internalizing receptor. After 70-min incubations, approximately 50% of RS7 had been internalized in MDA-MB-468 cells. An even higher percentage of RS7 was internalized in the Calu-3 cells.

Additional mAbs that bind to Trop2 have also been shown to be internalized. The mAb 2EF bound to Trop2 and was internalized [133]. A novel Fab antibody for Trop2 generated using the phage display technique showed strong internalization in breast cancer SKBR3 cells [134]. The anti-Trop2 mAb IR700 was conjugated to a photosensitizer and hence converted into a photoimmunotherapeutic (PIT) agent [135]. Nishimura et al. studied the internalization of the PIT agent by intravenously injecting mice bearing human pancreatic cancer PK-59 xenografts, followed by excision of the tumors and processing for confocal fluorescence imaging. The PIT agent was abundantly localized in the cytoplasm and was the first demonstration of a therapeutic anti-TROP2 antibody-based agent reaching the cytoplasm, most likely via endocytosis.

Interestingly, the MOv16 mAb, which binds to a different epitope than RS7, was able to stimulate a Ca^2+^ release in ovarian cancer and breast cancer OvCa-432 and MCF-7 cells, respectively [136]. RS7 only stimulated a Ca^2+^ release in the OvCa-432 cells, albeit more vigorously than MOv16. Other mAbs specific to Trop2, such as 162–46.2 and T16, induced minimal to no Ca^2+^ release. Although it was not explored, this study indicated that mAbs that bind Trop2 at different epitopes could induce different efficiencies for endocytosis, which is linked to the Ca^2+^ release and intracellular signaling.

Moon et al. were the first to develop humanized (h) RS7 as an ADC conjugated to the topoisomerase I inhibitor 7-ethyl-10hydroxycamptothecin (SN-38) via the novel crosslinker designated CL2 [137]. CL2 contains seven PEG groups and originally contained a Phe-Lys peptide for cathepsin recognition and cleavage. However, the phenylalanine was eliminated to simplify the chemical synthesis and it is thus known as CL2A and is the current linker used on SG. Cardillo et al. showed that challenging various tumor cell lines with hRS7-CL2A-SN-38 resulted in potent but varying IC_50_ values and no correlation between Trop2 expression levels [138]. Interestingly, the ratio of ADC:free SN-38 was lower in cells with relative increased Trop2, suggesting that increased expression corresponds to increased internalization and drug release/retention in tumor cells. Lung carcinoma Calu-3 cells showed the most internalization, whereas pancreatic cancer BxPC-3 cells showed 10.7-fold lower Trop2 expression and 2-fold reduced internalization. Unfortunately, an understanding of this relationship in the in vivo setting could not be attained. Although both Calu-3 and BXPC-3 xenografts were effectively eliminated, the injection sequence of hRS7-CL2A-SN-38 were different for both models and hence were not compared.

Therefore, numerous studies have demonstrated the importance of anti-Trop2 ADC/Ab-based therapeutic binding and subsequent endocytosis in a wide variety of tumor cell types as part of their effectiveness in killing tumors. In addition, endocytosis activity, but not receptor expression level, appears to be associated with increased tumor killing. There are also multiple mAbs against Trop2, and it appears that certain epitopes are more favorable for endocytosis.

### 3.11. BCMA

Of the billions of B cells that the normal immune system produces every day, only a few of these cells survive and further mature to provide humoral immunity, whereas the rest undergo cell death [139]. BCMA or CD269, also known as TNFR superfamily member 17, transduces signals for inducing B cell survival and proliferation [140]. BCMA has a molecular weight of only 20.2 kDa and its ligand-binding extracellular region has an ‘arm chair’ conformation [PDB 1OQD] and is composed of a single six-CRD [141] (Figure 1). Apart from multiple myeloma, BCMA is expressed in a number of hematologic malignancies, such as Hodgkin and non-Hodgkin lymphomas [142,143,144,145,146]. In multiple myeloma, BCMA is almost exclusively expressed on plasmablasts and plasma cells and is weakly detectable on some memory B cells committed to plasma cell differentiation and on plasmacytoid dendritic cells [147]. BCMA is also undetectable in naïve B cells, hematopoietic stem cells, and in normal non-hematologic tissues, except for some organs such as the testis, trachea, and some portions of the gastrointestinal duct due to the presence of plasma cells [147]. BCMA is expressed on 80–100% of multiple myeloma cell lines and is elevated on malignant plasma cells, relative to normal plasma cells [148,149]. BCMA is constitutively activated in multiple myeloma cells and induces growth and survival [150].

Surprisingly, there is minimal information on the precise endocytic pathway utilized by BCMA. Huang et al. investigated the effect of *N*-glycosylation on the function of BCMA [151]. It was discovered that various glycosylation forms occurred on residue Asp42. Specifically, when sialic-acid-based glycosylation occurred, BCMA’s pro-survival effects were suppressed due to interference with binding to its natural ligand, a proliferation-inducing ligand known as APRIL. In relation to endocytosis, sialylation is a regulatory function, as it induces BCMA to undergo endocytosis as a mechanism to enable non-sialylated BCMA at the cell surface to bind APRIL and initiate pro-survival signaling cascades. The authors surmised that because BCMA undergoes only a single glycosylation, it is probable that BCMA bind to siglecs, which readily recognize sialic acids [63] and utilize CME, as in the case of CD22, as previously described (Figure 1).

BCMA is a member of the TNFR family and information is available on its utilized endocytosis pathways. In general, TNFR members are categorized according to the presence or absence of a death domain [152]. BCMA does not contain a death domain. TNFR2, which also does not contain a death domain, utilizes CME [153]. In contrast, TNFR1, which does contain a death domain, utilizes caveolin-dependent endocytosis [154].

In relation to ADC targeting, patient serum levels of soluble BCMA correlate with disease status, response to therapy, and overall survival [155,156]. BCMA is constitutively shed from the plasma membrane by the intramembranous protease γ-secretase [157]. Soluble BCMA has been shown to interfere with anti-BCMA therapies by reducing the levels of active drug that reach multiple myeloma cells [155]. However, the H929 cell line, which expressed the highest relative amounts of cell surface and soluble BCMA, displayed the highest sensitivity to an investigational ADC [158]. This indicates that BCMA cell surface levels might have a greater impact on efficacy than the soluble BCMA concentration. Unfortunately, indirect evidence is all the information available at present regarding the endocytic mechanistic activities of BCMA.

### 3.12. ADC/Ab-BCMA Endocytosis

Endocytic pathways in relation to antibody or ADC internalization have yet to be reported. We have surmised potential endocytosis and intracellular sorting processes for ADCs based on evidence from findings with anti-BCMA fusion toxins.

Interestingly and potentially relevant for ADC efficacy, the initial biochemical characterization of BCMA protein expression on plasmacytic cells showed no expression at detectable levels on the cell surface, despite the fact that there was a high level of expressed protein [159]. BCMA was present exclusively in the Golgi apparatus. Fortunately, BCMA was indeed expressed on the cell surface of primary human B cells [140,160]. However, the level of expression in normal B cells and multiple myeloma cells was not reported until recently. Utilizing a novel anti-BCMA ADC (HDP-101), Figueroa et al. determined that the level of antibody-bound molecules against BCMA per cell ranged from 1171–8987 in the multiple myeloma cell lines H929, INA-6, U266, MM.1S-Luc, RPMI-8226, MM.1S, LP-1, OPM-2, SKMM.1, and L363 [158]. Importantly, this finding was in agreement with the number of bound antibodies per cell (1170–8987) measured in bone marrow plasma cells from patients with newly diagnosed or relapsed multiple myeloma. ADC HDP-101 is conjugated to the RNA polymerase II inhibitor α-amanitin. Upon binding to BCMA, ≥60% of bound HDP-101 was internalized after 2 h in multiple myeloma cell lines. This was determined to be due to rapid internalization. Although there was no correlation between BCMA cell surface density and sensitivity for HDP-101, the relatively markedly reduced expression at the cell surface demonstrates the importance of efficient endocytosis to ADC efficacy.

Kinneer et al. evaluated the anti-BCMA ADCs generated with enzymatically cleavable or non-cleavable linkers in combination with the warheads DM1, pyrrolobenzodiazepine (PBD), and MMAF and examined their performance in multiple myeloma cells in the absence and presence of protein lysosomal transporter solute carrier family 46 member 3 (SLC46A3) [161]. SLC46A3 is responsible for the efflux of the T-DM1 metabolite Lys-SMCC-DM1 out of lysosomes into the cytoplasm and its absence has been observed in T-DM1-resistant cell lines [162,163]. There was a wide range of SLC46A3 expression in multiple primary myeloma cells, including several samples with undetectable levels. Again, the expression of BCMA varied across samples, but approximately 90% expressed very low levels (as compared to the level on cell line H929, which has a modest BCMA cell surface density). The study revealed that SLC46A3-negative multiple myeloma cells were resistant to BCMA-targeting ADCs with DM1 and PBD, but were sensitive to those with MMAF. This study demonstrated that ADCs are indeed able to reach lysosomes and are processed, and their metabolites are either efficiently or inefficiently transported via efflux into the cytoplasm.

Studies with the fusion toxin consisting of the BCMA ligand B-lymphocyte stimulator (BLyS) and gelonin (BLyS-Gel) have elucidated endocytic pathways for BCMA. Lyu et al. evaluated whether BLyS-Gel cellular internalization involved an actin-dependent endocytic pathway [164]. Endocytosis was not affected by treatment with latrunculin A, a drug that sequesters actin monomers. Dose–response curves of BLyS-Gel with and without latrunculin A pretreatment were superimposable and therefore suggested that internalization did not involve an actin-cytoskeleton-dependent endocytic mechanism. Nonetheless, Luster et al. reported that BLyS-Gel was localized to lysosomes after internalization [165]. They showed that when cells were treated with chloroquine, a drug that accumulates in acidic endosomes/lysosomes, leading to their rupture, the fusion toxins were more efficacious. This suggested that BCMA-mediated endocytosis does indeed sort cargoes to lysosomes.

In summary, the endocytic machinery for BCMA internalization is unknown. Although there is no current evidence for it, there is an intriguing possibility that BCMA undergoes efficient internalization and sorting to lysosomes. Furthermore, multiple myeloma cells may release intracellular BCMA at the cell surface as part of the dysregulated cellular proliferation. Despite the fact that BCMA is modestly expressed on the cell surface, data showing that ADCs are indeed effective therefore makes BCMA a highly intriguing target. More importantly, BCMA endocytosis appears to be a major factor in ADC effectiveness.

### 3.13. HER2 and CME

Clinically, the anti-tumor activities of anti-HER2 (often referred to as ErbB2) therapies, with trastuzumab being the most recognized, are attributed to more than a single mechanism of action. Nonetheless, endocytosis and subsequent receptor downregulation likely plays a major role in its anti-tumor efficacy. In addition, the intracellular accumulation of the major cytotoxic metabolite of T-DM1 (Lys-SMCC-DM1) is directly correlated to therapeutic efficacy [8,166]. HER2 is a 185-kDa transmembrane glycoprotein and is a member of the EGFR family [167]. Amplification of the HER2/neu gene is a known driver of human malignancies and metastasis [168,169]. Owing to its role in cancer, HER2 has been pursued as a therapeutic target for decades. Trastuzumab is currently used as a highly effective frontline treatment for patients with HER2-positive breast cancer. As a result, HER2 has been a target for ADCs. Currently, T-DM1 and T-DXT are approved for use for patients with HER2-positive metastatic breast cancer who have received prior trastuzumab and chemotherapy. In addition, T-DXT is approved for patients who have received prior T-DM1 treatment. Thus, with the approval of two HER2-targeting ADCs, HER2 internalization and sorting to lysosomes must be very efficient.

However, HER2 is typically considered to be stable at the cell surface and when endocytosis does occur it is rapidly recycled back to the plasma membrane. HER2 was shown to be resistant to internalization in several tumor cell lines (SKBR3 [170,171,172,173,174,175,176], MDAMB-134 [177], T47D [178], BT474 [173], and HEp2 [176]) and models (HER2-transfected PAE cells, [173] and murine fibroblasts [177], and 3T3 [179] cells). Other studies have shown in 184A1 human mammary epithelial cells [180], SKBR3 [181], MCF7 [181], and BT474 [181] cells that HER2 does undergo endocytosis, albeit slowly, and is efficiently recycled back to the cell surface. Since no ligand that binds to HER2 has been identified, these findings have been a result of the fact that HER2 is the preferred heterodimerization partner of all other EGFR family members, which enhances their respective ligand binding and signaling properties [182]. In addition, HER2 stabilization is linked to its C-terminal intracellular domain and its interaction with HSP90 [177,179]. There has been much investigation into the interactions between HER2 and HSP90 and the mechanistic underpinnings for cell surface stabilization, and this is reviewed in [183]. As an additional and significant obstacle in relation to HER2 endocytosis, several studies have been performed in which cells were challenged with HSP90 inhibitors such as geldanamycin, thereby inducing artificial internalization. Although geldanamycin-induced treatment has elucidated several important aspects of HER2 endocytosis and intracellular sorting, from an ADC targeting perspective, this does not reflect a natural environment of receptor binding and subsequent internalization.

Co-immunoprecipitation has been effectively used to show that HER2 directly binds to AP-2 [179,184]. Cortese et al. showed that dynasore caused a complete block of HER2 endocytosis in SKBR3 cells, either in the presence or in the absence of geldanamycin [172]. In addition, HER2 underwent constitutive endocytosis regardless of the presence of galdanamycin. However, HER2 overexpression was inversely correlated with the rate of CCP and CCV formation. Another study showed that upon EGF stimulation, HER2, in heterodimer formation, takes EGFR out of CCPs [185]. Pust et al. studied the interaction between HER2 and lipid-raft-associated proteins known as flotillins [171]. In SKBR3 cells, HER2 colocalized with flotillin-1 and flotillin-2. When the flotillins were depleted in the absence of geldanamycin treatment, HER2 was efficiently internalized. Hence, HER2 is stabilized at the cell surface due to contributions from binding partners flotillins-1 and -2, HSP90, and its intracellular sequence.

#### 3.13.1. HER2 and Clathrin-Independent Endocytosis

##### HER2 and Caveolae-Mediated Endocytosis

The caveolin-binding motif φxφxxxxφ (φs represent the aromatic amino acids Trp, Phe, or Tyr) is typically found on caveolin-associated proteins [35]. Interestingly, the sequence WSYGVTIW has been identified in the intracellular kinase domain of HER2 [186]. Zhao et al. demonstrated that HER2 was localized in caveolin-3-positive cardiac myocytes stimulated with NRG or human glial growth factor 2, which are ligands for ErbB4. Ligand stimulation caused ErbB4 to translocate out of caveolae. In contrast, HER2 that had formed heterodimers with ErbB4 did not translocate out of caveolae. Zhou and Carpenter demonstrated that heregulin or betacellulin, but not EGF, promote the rapid translocation of HER2 to detergent-insoluble plasma membrane domains in T47D breast cancer cells [187]. It is unknown whether these detergent-insoluble domains were indeed caveolae, as the authors could not detect caveolin-1. Utilizing normal human fibroblasts, Mineo et al. demonstrated that HER2 was highly enriched in caveolin-1-positive plasma membrane fractions [188]. EGF stimulation induced the migration of EGFR out of caveolae but, once again, HER2 remained in these microdomains. Nagy et al. described the states of association of HER2 and its clustering in lipid rafts in SKBR3 cells [189]. HER2 was preferentially localized in lipid rafts and underwent endocytosis, but did not colocalize with caveolin-1. In addition, when the cells were treated with the B subunit of cholera toxin (CTX-B), which binds GM1, a raft-associated ganglioside [190], the caveloin-1-positive microdomains migrated laterally and formed larger clusters and HER2 was dislodged from lipid rafts and moved into the bulk lipid phase of the plasma membrane.

Pereira et al. evaluated multiple HER2-positive tumor cell lines, spanning gastric (N87, KATOIII), bladder (HT1197, UMUC14, UMUC3), breast (MDAMB231, MCF7, BT474, SKBR3), pancreatic (BxPc3, MiaPaCa, Suit2), and prostate (LnCap, Pc3) cancers [191]. HER2 and caveolin-1 expressions were inversely correlated. For example, SKBR3 cells, which have been the subject of debates about whether or not they express caveolae, did express caveolin-1 but at very low levels relative to its high HER2 expression. Although HER2 was present in caveolin-1-positive microdomains, it was also observed that tumor cells with high caveloin-1 expression contained HER2 clusters below the cell surface. In contrast, tumor cells with low caveolin-1 expression contained HER2 expression predominantly at the cell surface. When caveolin-1 expression was knocked down, particularly in cell lines with high expression, the level of HER2 and its retention time at the cell surface was increased. Conversely, forced overexpression of caveloin-1 promoted the loss of HER2 at the cell surface. The same occurred upon depletion of cholesterol using methyl-β-cyclodextrin and filipin. As a second example, when N87 cells were incubated with radiolabeled trastuzumab and cell surface or internalized radioactivity was measured, the studies revealed that when caveolin-1 was knocked down there was a marked increase of radioactivity at the cell surface and a reduction of intracellular radioactivity. As a third example, Lovostatin, a commercially available drug for the treatment of individuals with high blood cholesterol levels, was evaluated as a pharmacological modulator of caveolin-1 through the disruption of cholesterol-rich microdomains. Mice bearing bilateral xenografts of UMUC14 bladder and N87 gastric cancers were utilized to determine the impact of lovastatin-mediated caveolin-1 depletion on trastuzumab efficacy. Lovastatin was injected directly into one tumor, whereas the other tumor received no treatment. Mice were then intravenously injected with radiolabeled trastuzumab. Tumors treated with lovastatin showed >2-fold uptake of radiolabeled trastuzumab relative to untreated tumors. The increased tumor uptake occurred for both gastric and bladder cancer models. Exploiting this system, trastuzumab was able to increase the tumor killing of N87, BT474, and gastric primary-derived xenografts. This was convincing evidence that HER2 utilizes caveolae-mediated endocytosis in cancer and that the increased expression is due to the repression of caveolin-1, which enables the increased accumulation of trastuzumab in targeted tumors. However, the increased trastuzumab tumor accumulation (and improved tumor killing) was not due to increased HER2 caveolin-1-mediated endocytosis.

Taken together, these studies demonstrate that HER2 is present in lipid raft microdomains that most likely include caveolae and is internalized from these domains. Hommelgaard et al. demonstrated that HER2 is associated with membrane protrusions that did not contain CCPs [170]. Since the plasma membrane is highly dynamic and microdomains migrate, if this occurs, HER2 does not migrate with these domains, implying that the interaction between HER2 and caveolae could be transient in certain cell types. However, there is also a strong link between caveolae and HER2 that most likely occurs in cells with increased caveolin-1 expression but reduced HER2 levels. The balance between receptor expression at the cell surface and endocytosis and the potential impact on ADC effectiveness remain to be elucidated.

##### HER2 and CLIC/GEEC Endocytosis

Barr et al. demonstrated that HER2 can utilize an endocytic pathway similarly to CLIC/GEEC in SKBR3 cells treated with geldanamycin [192]. In geldanamycin-treated cells challenged with chlorpromazine, a cationic amphiphile that inhibits CME [193], chloropromazine efficiently inhibited the endocytosis of the transferrin receptor, which utilizes CME, but did not block HER2 internalization. Furthermore, HER2 did not colocalize with markers of the CME pathway. Interestingly, the structures containing internalized HER2 had a tubular morphology. Eps15 is a scaffold protein required for CME [194] and, as previously mentioned, dynamins mediate vesicle scission for both CME and caveolae-mediated endocytosis. Geldanamycin-treated SKBR3 cells expressing dominant-negative forms of Eps15 and dynamin-1 still had no effect on HER2 endocytosis. HER2 endocytosis did colocalize with PLAP and FluoroRuby dextran, both markers of CLIC/GEEC [192]. Once internalized, HER2 is transported to early and late endosomes, and then to lysosomes for degradation. In COS7 cells, which were transiently transfected with HER2 and GRAF1, which is a specific regulator of the CLIC/GEEC pathway [195], HER2 did not colocalize with GRAF1.

These various findings reveal several important features. First, HER2 endocytosis is promiscuous but a caveolae-mediated endocytic pathway appears to be more frequently utilized (Figure 1). HER2 endocytosis has largely been evaluated in SKBR3 cells and in the presence of HSP90 inhibitors. Whichever endocytic pathway HER2 utilizes is most likely an inefficient one, in which sorting to recycling endosomes dominates over lysosomal sorting.

### 3.14. Ab/ADC-HER2 Endocytosis

It is important to note that as studies were performed in SKBR3 or non-breast cancer model cells for evaluating the internalization and subsequent sorting to lysosomes. It is necessary for the model systems for ADC targeting to be conducted in tumor cells, and the HER2 expression level is also relevant. For example, T-DM1 efficacy is dependent on the level of expression of HER2 on cancer cells and it is known that patients who express HER2, defined by an IHC score of 3+, exhibit more frequent responses than patients with reduced levels [196,197]. Recently, HER2 endocytosis and its relationship with Ab/T-DM1 efficacy was reviewed in [8]. Briefly, cell origin, as well as the receptor expression level, greatly influence internalization and lysosomal degradation efficiency. Targeting HER2/ErbB heterodimers can enhance antibody internalization and is a feature observed in cells with reduced HER2 expression. Increased HER2 density and internalization occurs in areas of the membrane undergoing rapid reorganization of the plasma membrane. In summary, there is at least a sufficient fraction of T-DM1 that is able to internalize and be processed in lysosomes for the DM1 release. For cells with high HER2 expression, poor internalization may be compensated for by the excess amounts of HER2 at the cell surface, meaning that sufficient receptors are still internalized. For cells with low HER2 expression, the better internalization may compensate for reduced amounts of HER2 at the cell surface. Approaches aiming to improve HER2-specific Ab/ADC internalization were also reviewed in [8]. The most successful approach is receptor crosslinking with antibody cocktails or via bispecific and biparatopic Abs/ADCs. We briefly describe these approaches in detail and include our most up-to-date findings.

Recently, biparatopic engineering technology has made an important advancement in understanding HER2 endocytosis and, more importantly, how it impacts ADC therapeutic efficacy. Li et al. developed a biparatopic tetravalent ADC composed of trastuzumab and mAb 39S binding domains that target non-overlapping epitopes in domains IV and II, respectively [198]. The ADC was conjugated to four molecules of the tubulysin derivative AZ13599185 and the complete ADC is known as MEDI4276. The biparatopic ADC induced extensive HER2 crosslinking that resulted in increased internalization, lysosomal localization, and HER2 degradation. As a result, MEDI4276 was highly effective in tumor models representing different breast cancer patient subpopulations that varied in HER2 expression due to the significant improvement in internalization.

Cheng et al. examined the molecular mechanism and critical determinants that promoted accelerated HER2 endocytosis [199]. High BT474- and SKBR3- and low MCF7-expressing cells were treated with Pitstop 2 or Dyngo 4a, which are CME- and dynamin-selective inhibitors, respectively. Both MEDI4276 and trastuzumab had their internalization blocked in the presence of each inhibitor. MEDI4276 had an internalization half-life of 30 min. In contrast, the trastuzumab half-life was 120 min. Accordingly, >50% of HER2 molecules were degraded within 2 h of MEDI4276 treatment. MEDI4276 stayed bound to HER2, whereas trastuzumab and 39S dissociated. This supports previous studies that showed that the fraction of bound trastuzumab that dissociates from HER2 after internalization is recycled back to the cell surface [181]. In addition, this new finding suggests that ADCs destined for lysosomal degradation must remain bound to HER2. In cells with low surface expression of HER2, the receptors are sparsely located, and the crosslinking-induced effect of MEDI4276 was not as robust, as there were fewer HER2 clustering complexes relative to cells with high HER2 expression. This resulted in slower endocytosis kinetics and cytotoxicity.

In summary, this study revealed that inducing HER2 rapid internalization is most likely a key parameter in improving ADC efficacy. Crosslinking HER2 has been proven to be an effective preclinical approach to induce rapid endocytosis [8] and has shown efficacy in patients [200]. In this scenario, there is a critical, unknown receptor number and density threshold on the tumor cell surface, which is required for efficient endocytosis. This is also beneficial for determining toxicity effects. As HER2 is expressed on heart tissue, however, the expression of HER2 is low.

Kang et al. recently determined that the potency of anti-HER2 ADCs could be improved by substantially reducing their binding affinity for HER2 at acidic endosomal pH relative to near-neutral pH, in a process termed ‘acid-switching’ [201]. The rationale behind this approach is most likely based on what was previously described above, namely, that antibodies bound to internalizing HER2 can dissociate from one another due to the acidic endosomal environment, which leads to HER2 recycling back to the cell surface while the antibody is sorted to lysosomes [181]. Pertuzumab was viewed as an attractive mAb for engineering due to the fact that its affinity for HER2 is approximately 10-fold stronger at pH 7.4 than at pH 6.0. In contrast, trastuzumab has similar binding affinities for HER2 across the pH range 6.0–7.4. The residues in the complimentary determining regions (CDRs) of pertuzumab that directly interact or are in close proximity with HER2 were mutated. The CDRs were randomly mutated further by means of the phage display technique, followed by panning to isolate acid-switched variants. Two candidates were identified based on the mutations Y55H/S103H and G57E/S55H, termed ‘YS’ and ‘SG’, respectively. Both variants showed similar binding affinities to HER2 at pH 7.4, but YS and SG had 2.5- and 6.4-fold weaker affinity relative to pertuzumab at pH 6.5. SG and YS were conjugated to vc-PAB-MMAE. In highly HER2-expressing MDA-MB-453 cells, SG and YS colocalized with lysosomes, whereas pertuzumab exhibited minimal colocalization. Measuring the intracellular accumulation of the delivered MMAE revealed that SG and YS increased the level of cellular MMAE up to 3.5- and nearly 2-fold in MDA-MB-453 and JIMT-1 cells, respectively. This resulted in a cytotoxic potency improvement by approximately 2-logs in MDA-MB-453 cells. However, there was a marginal cytotoxic improvement in JIMT-1 cells. In other HER2-positive cell lines (MDA-MB-468, SKBR3, and HCC1954) there was no cytotoxic improvement. Mice bearing small-volume-sized xenografts (determined by the short duration between the commencement of treatment after tumor cell implantation) were treated at a suboptimal dose of 2 mg/kg. In this tumor model, both SG-MMAE and YS-MMAE significantly outperformed pertuzumab-MMAE and T-DM1 in the MDA-MB-453 tumor model. SG-MMAE (YS-MMAE not evaluated) also improved the killing of JIMT-1 xenografts. This proof-of-concept study demonstrated a novel approach aiming to exploit the endosomal sorting of bound antibodies to promote localization to lysosomes, offering a potential approach to overcoming the natural inefficient endocytosis of HER2.

### 3.15. Nectin-4

The nectin family of immunoglobulin-like transmembrane proteins comprises four members, nectin-1 to nectin-4. Nectin-4 is a type-I transmembrane 66 kDa protein and its main role is facilitating cell–cell contact [202]. Specifically, nectin-4, together with cadherins, form specific intercellular adhesive structures named adherens junctions [203]. Nectin-4 is attractive as an ADC target because studies have shown that it is overexpressed in several tumor types but nearly absent in normal adult tissues. Originally, Reymond et al. reported that nectin-4 was weakly expressed in the trachea among 23 normal human tissues tested [204]. More recently, Challita-Eid et al. showed that additional normal tissues such as the skin, bladder, salivary gland, esophagus, breast, and stomach also express weak-to-moderate levels of nectin-4 [205]. When evaluating nectin-4 expression in cell lines, Fabre-Lafay et al. demonstrated that relative to immortalized normal endothelial and hematopoietic cells, which did not show expression, only 1/4 prostate and bladder cancer cells overexpressed nectin-4 [206]. For ovarian and breast tumor cell lines, nectin-4 was overexpressed in 50% (4/8) and 68% (23/34), respectively. Nectin-4 expression by IHC on >760 breast carcinoma samples revealed that nectin-4 was more prominently expressed in ductal breast cancer where strong cytoplasmic expression was observed. Interestingly, nectin-4 is also cleaved by ADAM-17 and evaluation of serum nectin-4 levels at diagnosis, correlated with the presence of multiple breast tumor metastases [206]. In 2394 patient tumor specimens, nectin-4 was overexpressed in 60% and 53% of bladder and breast tumor tissues, respectively [205].

No information was found on natural ligands or mAb/ADC endocytosis in complex with nectin-4 and hence we searched for the pathogen binding of nectin-4 internalization to draw parallels. Nectin-4 is also a receptor for the measles viruses [207]. Due to the excellent preferential tumor overexpression of nectin-4, its endocytosis properties have been evaluated as a potential target for oncolytic viruses and, as a result, molecular players involved in receptor-mediated endocytosis have been elucidated and can be beneficial for the understanding ADC mechanisms of action.

Delpeut et al. showed that the measles virus entered MCF7 and HTB-20 breast and DLD-1 colorectal cancer cells via macropinocytosis (Figure 1) [208]. PAK1 was required for viral entry. In contrast, the dynamin-inhibitor Dynasore did not have an effect on virus entry. In addition, cells expressing dominant-negative caveolin did not abrogate viral endocytosis. Disruption of actin filaments with cytochalasin D blocked endocytosis. Interestingly, the endosomal acidification inhibitor bafilomycin A1, which increases late-stage endosomal pH, did not affect viral endocytosis. This is most likely due to the fact that the measles virus escapes from early and recycling endosomes.

Based on these indirect studies, nectin-4 exhibits the robust endocytosis activities that a viral receptor would need. In addition, at least partially, it appears to traffic to lysosomes. These are the essential qualities of an attractive target for an ADC. Further studies will be required in order to understand the endocytosis activity of nectin-4 as it relates to ADCs.

## 4. Dysregulated Endocytosis and ADC Resistance

### 4.1. Endophilins

A functional endocytic vesicle requires cargo recruitment adaptors, membrane curvature effectors and a membrane scission component. Endophilins have all these characteristics in one protein. Endophilins are well-characterized BAR-containing proteins that also contain an SH3 domain that is critical for their function. The endophilin BAR domains induce and stabilize membrane curvature and are able to ‘sense’ and bind to already curved membranes [209]. In addition, endophilins promote dynamin recruitment for narrowing of the endocytic neck and ultimately scission and formation of the fully-formed intracellular vesicle [210]. As they are such important players in the endocytic process, it is not surprising that endophilins are associated with both CME and clathrin-independent endocytosis [210,211,212].

Noting that endophilins are important in the endocytosis process, there are numerous findings associating these proteins with cancer. Endophilin A1 (Endo A1) has shown decreased expression in several tumor types [209]. Endo A1 gene alterations are associated with poor prognosis in patients with breast cancer [213]. Endophilin B1 was found to be downregulated in lung carcinomas [214].

Endophilin 2 (Endo A2) is vital as it acts as a scaffold that tethers the membrane as actin pulls the budding endocytic vesicle downward to facilitate scission. When Endo A2 is overexpressed, ligand-induced endocytosis is increased [215]. Endo A2 has also been shown to direct a distinct, clathrin-independent endocytic pathway known as fast endophilin-mediated endocytosis (FEME) that acts on many classes of receptors [211,216].

Endo A2 dysregulation and its effects have a more pronounced role in endocytosis and ADC resistance. The myeloid-lymphoid leukemia (*MLL*) gene encodes a methyl transferase involved in normal gene regulation. However, the translocation of MLL frequently occurs and leads to genetic fusions and chimeric proteins associated with leukemias prevalent in infants of poor prognosis [209]. One such fusion is the MLL-Endo A2 chimeric protein. As a result of this fusion protein, normal Endo 2A levels are significantly reduced [217]. The reduced amount of normal Endo 2A required for endocytosis resulted in the attenuation of endocytosis-based downregulation of growth factor receptors. The inability of MLL-Endo A2-containing cells to downregulate the number of growth factor receptors resulted in over-activation of signaling cascades and tumor growth.

Endo A2 was also observed to be overexpressed in tumors and metastatic lymph nodes, with the highest expression in HER2-positive cases [218]. Increased expression of the Endo A2 gene (*Sh3gl1*) was associated with reduced relapse-free and overall survival. Lamellipodin is an adaptor protein that interacts with various scaffold proteins and actin and is crucial for various processes related to plasma membrane protrusions [219]. Lamellipodin also binds to Endo A2 for the facilitation of FEME. Lamellipodin gene (*RAPH1*) transcription was found to be increased and associated with significantly worse relapse-free survival in node-positive HER2-positive patients [218]. Notably, in certain cases, Endo A2 expression was similar to HER2 expression levels. This finding suggests that the increased presence of Endo A2 (to extreme levels comparable to HER2) increases the endocytosis of important cell surface receptors, including HER2, that increase the signaling and activation of important genes that promote cell migration and invasion.

Baldassare et al. reported that Endo A2 is a critical factor that promotes sensitivity to T-DM1 and that, when silenced, cells have reduced sensitivity [218]. The HER2-positive tumor cell lines SKBR3 and HCC1954 contained higher Endo A2 levels relative to the normal-like breast epithelial cell line MCF-10A. Endo A2 levels were reduced by ~85%. Silencing of Endo A2 led to HER2-positive tumors that were significantly less sensitive to T-DM1 and trastuzumab treatment relative to cells with normal Endo A2 levels. Interestingly, when cells were treated with the small molecule HER2 inhibitor lapatinib, there was no difference in cytotoxicity. These findings suggest that impaired endocytosis-mediated resistance is a major factor blocking the efficacy of T-DM1 and trastuzumab.

In summary, variable expression levels of endophilins are associated with tumor aggressiveness and poor patient outcomes. Low levels of Endo A2 may lead to T-DM1 resistance. Further efforts to stratify HER2-positive patients with endocytic protein markers such as endophilin may be required in order to avoid resistance to T-DM1.

### 4.2. Caveolins

Sung et al. reported a novel mechanism for trastuzumab ADCs based on caveolae-mediated endocytosis [220]. HER2-positive gastric cancer N87 cells were repetitively treated in vitro using a schema that mimicked the clinical dosing regimen of T-DM1. The resistant N87 cells showed an increased expression of caveolin-1 and transcript release factor (PTRF). PTRF is an essential protein used in caveolae-mediated endocytosis [221]. Internalization studies showed that T-DM1 colocalized with caveolin-1 in resistant but not in wild-type N87 cells. In addition, caveolin-1-positive intracellular vesicles had significantly reduced levels of LAMP-1 and contained neutral pH centers, as opposed to the acidic centers contained by lysosomes. As a result, various trastuzumab-ADCs based on noncleavable linkers and anti-tubulin payloads were 7–188-fold less potent in the resistant N87 cells.

Smith et al. reported on an investigational ADC specific for p97, a glycosylphosphatidyl inositol-linked receptor internalized via the caveolae pathway [222]. In melanoma cells, the ADC colocalized with the lysosome marker CD107a in sensitive cells. In contrast, in resistant cells the ADC colocalized with caveolin-1 and not with CD107b. Again, resistant cells were less sensitive compared to wild-type cells to ADC treatment.

## 5. Clinical Perspectives

Biopharmaceuticals targeting cell surface receptors are frequently subject to target-mediated clearance, resulting in nonlinear pharmacokinetic (PK) profiles. Tumor uptake efficiency is highly dependent on PK. Endocytosis of a biopharmaceutical upon binding to a cell surface receptor is the underlying cause of target-mediated drug clearance. ADC internalization kinetics most likely have a significant impact on efficacy and off-tumor side effects. Moreover, the internalization of receptors with expression as low as 5000 molecules per cell can affect PK and increased internalization has been associated with more pronounced receptor-mediated clearance [223]. The clinical implications include nonlinear PK when target receptors are not fully saturated, the administration of a non-ideal dose or frequency in order to maintain target engagement, and an economically unviable increase in drug price. Given the lengthy and costly ADC development process, the evaluation and selection of the optimal cancer receptor for targeting and the optimal mAb for ADC development is imperative in order to increase the probability of success.

Information on the clinical effects with respect to internalization is scarce and, consequently, at this time it is difficult to determine relationships between all or most vital variables (e.g., linker type, payload, internalization kinetics, receptor expression, solid/hematological tumor, etc.). In terms of endocytosis, target-mediated clearance can be tumor-specific or non-specific. Non-specific endocytosis and degradation involve target receptors expressed on healthy tissues. The level of expression and the rate of internalization relative to the tumor further complicate matters.

Deslandes analyzed 21 ADCs with reported information on PK parameters at multiple doses or information on dose ranges and dosing regimens to elucidate dose–clearance relationships [224]. GO displayed rapid clearance relative to other ADCs, and it was suggested that a possible reason was due to CD33-mediated internalization and the high rate of renewal that occurs in AML [224]. Originally, the recommended GO treatment regimen was a total of two 9 mg/m^2^ doses with 14 days between administrations. GO was withdrawn from the market in 2010 when its efficacy was found to be poor and its toxicity appeared excessive. The new dosing regimen of 3 mg/m^2^ on days 1, 4, and 7 allows for the safe administration of higher cumulative doses and significantly improved outcomes in patients with AML [225]. Importantly, the rationale for the administration of fractionated doses was based on the rapid endocytosis and re-expression of CD33. However, there was no general rule found for other ADCs, including the approved ADCs BV, T-DM1, InO, and PV. Some ADCs showed reduced blood clearance with increased doses, whereas other ADCs showed no dose-dependent changes in clearance.

Recently, Goldenberg and Sharkey rationalized a link between the excellent internalization of Trop2 and/or SG (as previously described in Section 3.9 and Section 3.10) [226]. SG is considered to be a product of a novel ADC development platform, one that utilizes a payload (SN-38) of lower potency than the ultratoxic agents (i.e., nanomolar as opposed to picomolar IC_50_ values) and is coupled to a pH-labile linker (CL2A) with moderate stability. The linker is essential as it enables SG to readily release its payload in a rapid fashion due to the high internalization rate of Trop2 and thereby accelerates the drug delivery to tumor cells. Interestingly, given the sensitivity of the CL2A linker to acidic pH, SG releases SN-38 over a period of about 3 days, and the authors surmise that some portion of the therapeutic activity of SG could result from bystander effects, due to the local release of the payload in the tumor microenvironment. This ultimately results in effective cytotoxicity of tumor cells. This demonstrates that endocytosis is vital but also that the interplay between the linker type and internalization is also important.

Therefore, the endocytic activities of targeted receptors on tumor cells, as well as normal cells, can significantly contribute to the unique dose-dependent PK profiles and dosing for individual ADCs. This is important as ADCs exhibit therapeutic indices that are dependent upon the cytotoxic differences between the tumor and normal cells. In the future, clear study designs should be established to determine the relationships between the effects of receptor endocytosis, PK parameters, and patient outcomes.

## 6. Conclusions

The analysis of the endocytosis of target receptors and the internalization of ADCs can greatly facilitate preclinical development, clinical translation, and patient treatment efficacy. It is apparent that the efficiency of endocytosis and subsequent routing to lysosomes (or other protease-present subcellular locations) is just as important as the preferential overexpression status of target receptors for ADCs and additional pertinent parameters such as PK. Although there are now nine approved ADCs and the field is experiencing a resurgence due to technical advancements in ADCs, there have been >100 ADCs in clinical trials at one time or another over the last 25 years, and most of those agents have failed [227]. Moreover, there are several times more ADCs in preclinical development. The determination of the drug dose–exposure effect relationship is a crucial part of ADC success and, consequently, the internalization is a vital part of any relationship and will be important for optimizing the dosing regimen to maximize the therapeutic index.

Interest in ADC endocytosis has increased with the discovery of additional overexpressed receptors on tumor cells and high-affinity-binding mAbs. Most investigational ADCs will demonstrate a minimum capability to exploit target receptor internalization processes as a means to justify the ADC concept for delivery payloads inside tumor cells. Despite this, there is still little understanding of the endocytosis of target receptors. Given the large number of differences between targets, ADC constructs, dosing regimens, and patient populations, dedicated streams of internalization and endosomal sorting processes—in parallel with traditional ADC development areas (conjugation, linker design, cytotoxicity, anti-tumor efficacy, PK, etc.)—will increase the understanding of ADCs and inform researchers about how best to develop their therapeutic potential. In addition, due to the vital importance of many core endocytic components and key adaptors, these proteins are likely to be found to be commonly mutated in cancer and this too will affect ADC endocytosis and efficacy. Lastly, as the field has evolved, and after decades of experience, each individual aspect of an ADC is likely to be vital to ADC efficacy, and measuring all of these components is important. In this review, endocytosis has been described as an important additional aspect.

## Figures and Tables

**Figure 1 pharmaceuticals-14-00674-f001:**
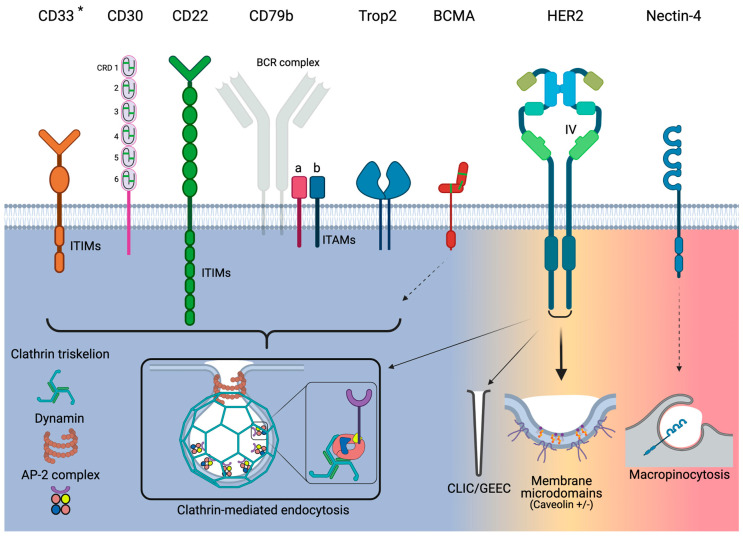
Overview of endocytosis pathways utilized by the target antigens for the currently approved ADCs. Narrow arrows indicate minor utilization by HER2. Dashed arrows indicate that only indirect evidence exists. * Although CD33 utilizes clathrin-mediated endocytosis, it acts independently of AP-2. Green bars for CD30 and BCMA represent disulfide bonds.

**Table 1 pharmaceuticals-14-00674-t001:** Currently approved ADCs and indications.

ADC	Target	Indications and Usage ^1^
Gemtuzumab ozogamicin (GO)	CD33	● Newly diagnosed and relapsed and refractory acute myeloid leukemia (AML) ^2^
Brentuximab vedotin (BV)	CD30	● Hodgkin lymphoma after failure of autologous stem cell transplant (ASCT) or after failure of at least two prior multi-agent chemotherapy regimens in patients who are not eligible for ASCT ● Systemic anaplastic large cell lymphoma (ALCL) after failure of at least one prior multi-agent chemotherapy regimen
Inotuzumab ozogamicin (InO)	CD22	● Relapsed or refractory B cell precursor acute lymphoblastic lymphoma (ALL)
Polatuzumab vedotin (PV)	CD79b	● In combination with bendamustine and rituximab for relapsed or refractory diffuse large B cell lymphoma after at least two prior treatments
Sacituzumab govitecan (SG)	Trop2	● Triple-negative breast cancer after at least two prior therapies for metastatic disease
Trastuzumab emtansine (T-DM1)	HER2	● Metastatic breast cancer patients who previously received trastuzumab and a taxane ● Adjuvant treatment in early breast cancer with residual invasive disease after neoadjuvant taxane- and trastuzumab-based treatment
Trastuzumab deruxtecan (T-DXd)	HER2	● Unresectable or metastatic breast cancer after two or more prior anti-HER2-based regimens in the metastatic setting
Enfortumab vedotin (EV)	Nectin-4	● Locally advanced or metastatic urothelial cancer after a PD-1 ^3^ or PD-L1 inhibitor, and platinum-containing chemotherapy in the neoadjuvant/adjuvant, locally, advanced or metastatic setting
Belantamab mafodotin (BM)	BCMA	● Relapsed or refractory multiple myeloma after at least four prior therapies including an anti-CD38 mAb, a proteasome inhibitor, and an immunomodulatory agent

^1^ For adults. ^2^ In addition, for relapsed or refractory AML in adults and in pediatric patients 2-years and older; ^3^ Programmed death receptor-1 and programmed death-ligand 1.

**Table 2 pharmaceuticals-14-00674-t002:** Summary of endocytosis activities for the receptors for the currently approved ADCs.

Receptor	Pathway	Activity	Association with ADC Efficacy/Resistance
CD33	CME	Poor	● AML patients who do not respond to GO have been linked to poor receptor internalization
CD30	CME	Poor	● Undergoes significant shedding from cell surface
CD22	CME	Good	● Fast endocytosis activates intracellular pools, which replenish the level of CD22 expression
CD79b	CME	Good	● Due to rapid internalization and trafficking to lysosomes, patients will most likely respond to PV treatment
Trop2	CME	Good	● Strong preclinical data link internalization to efficacy
BCMA	Insufficient information	Good	● Insufficient information
HER2	Clathrin-independent (caveolae +/−)	Poor	● Poor internalization linked with poor clinical outcomes ● Dysregulation of the endocytotic machinery has been linked to resistance in preclinical models ● Novel strategies such as induced HER2 crosslinking to improve endocytosis are currently in clinical testing
Nectin-4	Macropinocytosis	Good	● Insufficient information

## Data Availability

Not applicable.

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
