# Peer review of "Impact of Endocytosis Mechanisms for the Receptors Targeted by the Currently Approved Antibody-Drug Conjugates (ADCs)—A Necessity for Future ADC Research and Development"

_pharmaceuticals, 2021, doi:10.3390/ph14070674_

Round 1
Reviewer 1 Report
The manuscript submitted by Hammood et al. entitled "Impact of Endocytosis Mechanisms for the Receptors Targeted by the Currently Approved ADCs – A Necessity for Future ADC Research and Development" deeply revises the effects of receptor trafficking pathways in the efficacy of the biological-based therapies. The molecular mechanisms are evaluated in different cancer types, since most of them can be disrupted. Although, some reviews were recently published in this field (e.g. https://doi.org/10.1186/s13045-021-01035-z; https://doi.org/10.1016/j.apsb.2020.04.012; https://doi.org/10.1016/j.ddtec.2020.07.002), this review reports a different clinical landscape which is highly relevant for the field.
Author Response
Please see attached response to Reviewer 1

Reviewer 2 Report
Overall comments
The effectiveness of cell targeted therapies such as antibody-drug-conjugates (ADCs) rely on internalization and endocytosis via target receptors. In cancer cells, some of the endocytic trafficking pathways may be dysfunctional. The last years have witnessed significant advances in ADC technology that has led to a number of FDA approvals. The authors state that the impact of dysregulated internalization processes of ADC targets and response rates or resistance have not been well studied. In this review, the authors we describe what is known about the internalization efficiency and relevant intracellular sorting activities for the currently approved ADCs and how aberrant endocytic processes might be linked to preclinical ADC resistance mechanisms with implications for therapeutic effectiveness in the clinic. The authors claim the review provides a better understanding of the principles of receptor endocytosis and application for ADCs in cancer therapy.
Specific comments
The authors had performed a commendable task in reviewing the literature and in highlighting the importance of including knowledge of the endocytotic pathways as part of the criteria for selection of candidate antibodies. Several points are suggested.
- Certain aspects regarding the uptake pathways that ADC pathways utilized are missing. One of these is the fact that at least with the earlier ADCs, including CD33 and CD30, antibodies were selected for their apparent target selectivity, not for their uptake or trafficking capabilities. Second, uptake efficiencies have been demonstrated in several studies to also depend of the pharmacokinetics of the antibody, not just its cell biology.
- Given the evidence of some dysregulation of endocytotic pathways in cancer cells, some discussion should be added as to the possibility of rationally targeting receptors that utilize particular endocytotic pathways to afford efficient entry and delivery of drug cargo.
- As an extension to 2), a table would help the reader associate the efficacy of approved ADCs with their endocytotic pathways in normal and cancer cells.
